



MOSAIC (Modern Ocean Sediment Archive and Inventory of Carbon):
A (radio)carbon-centric database for seafloor surficial sediments
Tessa Sophia van der Voort[1, †], Thomas M. Blattmann[1, ††], Muhammed Usman[1, †††], Daniel
Montluçon[1], Thomas Loeffler[1], Maria Luisa Tavagna[1], Nicolas Gruber[2], and Timothy Ian
Eglinton[1]
[1]*Department of Earth Sciences, Geological Institute, ETH Zürich, Sonneggstrasse 5, 8092*
*Zürich, Switzerland*
[2]*Department of Environmental System Sciences, Institute of Biogeochemistry and Pollutant*
*Dynamics, ETH Zürich, Universitätstrasse 16, 8092 Zürich, Switzerland*
[†] New address: Campus Fryslân, University of Groningen, Wirdumerdijk 34, Leeuwarden
[††] New address: Biogeochemistry Research Center, Japan Agency for Marine-Earth Science
and Technology (JAMSTEC), Yokosuka, Japan.
[†††] New address: Dept. of physical and environmental Sciences, University of Toronto M1CA4
Ontario, Canada
**Journal:** ESSD- Earth System Science Data
**Key points paper:**
(1) Paper presents global database for marine surficial sediments
(2) Database has a user-friendly interactive app with downloadable data
(3) Provides a new platform to answer key questions in biogeochemistry
**Key words:**
Ocean Sediments, Organic Carbon, Radiocarbon, [13]C, Carbon Sequestration, MOSAIC,
Database



Abstract
Mapping the biogeochemical characteristics of surficial ocean sediments is crucial for
advancing our understanding of global element cycling, as well as for assessment of the
potential footprint of environmental change. Despite their importance as long-term repositories
for biogenic materials produced in the ocean and delivered from the continents,
biogeochemical signatures in ocean sediments remain poorly delineated. Here, we introduce
MOSAIC (Modern Ocean Sediment Archive and Inventory of Carbon; DOI:
https://doi.org/10.5168/mosaic019.1, mosaic.ethz.ch, Van der Voort et al., 2019), a
(radio)carbon-centric database that seeks to address this information void. The goal of this
nascent database is to provide a platform for development of regional to global-scale
perspectives on the source, abundance and composition of organic matter in marine surface
sediments, and to explore links between spatial variability in these characteristics and
biological and depositional processes. The database has a continental margin-centric focus
given both the importance and complexity of continental margins as sites of organic matter
burial. It places emphasis on radiocarbon as an underutilized yet powerful tracer and
chronometer of carbon cycle processes, and with a view to complementing radiocarbon
databases for other earth system compartments. The database infrastructure and interactive
web-application are openly accessible and designed to facilitate further expansion of the
database. Examples are presented to illustrate large-scale variabilities in bulk carbon properties
that emerge from the present data compilation.




## 1. Introduction

Oceans sediments constitute the largest and ultimate long-term global organic carbon (OC)
sink (Hedges and Keil, 1995), and serve as a key interface between short- and long-term
components of the global carbon cycle (Galvez et al., 2020). Assessments of the distribution
and composition of OC in ocean sediments are crucial for constraining carbon burial fluxes,
the role of ocean sediments in global biogeochemical cycles, and in interpretation of
sedimentary records. Constraining the magnitude of carbon stocks, as well as delineating the
sources, pathways and timescales of carbon transfer between different reservoirs (e.g.,
atmosphere, oceanic water column, continents) comprise essential challenges. In this regard,
radiocarbon provides key information on carbon sources and temporal dynamics of carbon
exchange. The half-life of radiocarbon is compatible with assessments of carbon turnover and
transport times within and between different compartments of the carbon cycle, while also
serving to delineate shorter-term (< 50 kyr) and longer-term (> 50 kyr) cycles. Moreover, the
advent of nuclear weapons testing in the mid 20th century serves as a time marker for the onset
of the Anthropocene (Turney et al., 2018), and a tracer for carbon that has recently been in
communication with the atmosphere. With on-going dilution of this atmospheric "bomb spike"
with radiocarbon-free carbon dioxide from the combustion of fossil fuels (Graven, 2015; Suess,
1955), radiocarbon serves a particularly sensitive sentinel of carbon cycle change.

Radiocarbon databases or data collections have been established for the atmosphere (e.g.
University Heidelberg Radiocarbon Laboratory, 2020), ocean waters (Global Data Analysis
Project (GLODAP), Key et al., 2004), and most recently soils (ISRaD; Lawrence et al., 2020)
, with tree-rings, corals and other annually-resolved archives providing information on
historical variations in $^{14}$C in the atmosphere and surface reservoirs (Friedrich et al., 2020;
Reimer et al., 2009). At present, no such radiocarbon database exists for OC residing in ocean
sediments. As a sensitive tracer of carbon sources and carbon cycle perturbations, there is a
clear imperative to fill this information void given that on-going anthropogenic activities
directly and indirectly influence ocean sediment and resident OC stocks (Bauer et al., 2013;
Breitburg et al., 2018; Ciais et al., 2013; Keil, 2017; Regnier et al., 2013; Syvitski et al., 2003).
Materials accumulating in modern ocean sediments also provide a crucial window into how
on-going processes that are observable through direct instrumental measurements and remote
sensing data manifest themselves in the sedimentary record.




Over 85% of OC burial in the modern oceans occurs on continental margins, with deltaic, fjord
and other shelf and slope depositional settings constituting localized hotspots for carbon burial
(Bianchi et al., 2018; Hedges and Keil, 1995) . As the interface between land and ocean,
continental margins comprise a key juncture in the carbon cycle (Bianchi et al., 2018), provide
crucial habitats for unique marine ecosystems (Levin and Sibuet, 2012), support a major
fraction of the worlds fisheries (Worm et al., 2006), and participate in exchange processes with
the interior ocean (Dunne et al., 2007; Jahnke, 1996; Rowe et al., 1994). These ocean settings
and their underlying sediments are also amongst those most vulnerable to change (Keil, 2017)
through direct perturbations such as contaminant and nutrient discharge from land, loci of
intense resource extraction such as bottom trawling (Pusceddu et al., 2014) and mineral and
hydrocarbon recovery (e.g., Chanton et al., 2015), as well as indirect effects such as ocean
warming (Roemmich et al., 2012), acidification (Feely et al., 2008; Orr et al., 2005) and local
or large-scale deoxygenation (Diaz and Rosenberg, 2008; Keeling et al., 2010). Such influences
may change not only the amount of carbon sequestered in marine sediments but also its
character, with radiocarbon serving as a key metric to detect such change.

At present, an information gap exists between the numerous in-depth biogeochemical
investigations of carbon burial focused on geographically-localized regions (e.g. Bao et al.,
2016; Bianchi, 2011; Castanha et al., 2008; Kao et al., 2014; Schmidt et al., 2010; Schreiner et
al., 2013) and global-scale syntheses that draw upon large suites of bulk OC concentration
measurements but are limited in diversity of geochemical information (e.g. Atwood et al.,
2020; Premuzic et al., 1982; Seiter et al., 2004, 2005) and lack sedimentological context.
Consequently, current global-scale budgets and global-scale Earth System Models (ESMs) do
not resolve regional or small-scale variability (Bauer et al., 2013), and are limited by our
current understanding of variability in biogeochemical and sedimentary processes that
influence sedimentary organic matter composition and reactivity (Levin & Sibuet, 2012; Bao
et al., 2018; Arndt et al., 2013). Increasingly powerful Region Oceanic Model Systems
(ROMS) models (e.g., Gruber et al., 2012) and statistical methods for geospatial analysis (e.g.,
van der Voort et al., 2018; Atwood et al., 2020) hold the potential to utilize information from
local-scale studies and inform ESMs, but these require mining and collation of existing data
and merging this with new observations. Spatially-resolved datasets for marine sedimentary
OC are beginning to emerge (e.g. Inthorn et al., 2006; Schmidt et al., 2010), including
radiocarbon measurements (e.g., Bao et al., 2016; Bosman et al., 2020). The latter information



is likely to increase in availability with the advent of natural-abundance $^{14}$C measurement via
elemental analysis coupled with gas-accepting accelerator mass spectrometry (AMS) systems
(McIntyre et al., 2016; Wacker et al., 2010) that enable routine, high-throughput $^{14}$C
measurements.

Overall, there is a strong need to synthesize information related to not only OC content, but
also its composition and depositional context, from separate region-based studies. Merging of
this information to provide pan-continental margin ocean floor data resources would enable
development of robust budgets and detection in changes in the magnitude or nature of carbon
stocks. In addition to the content and radiocarbon characteristics of OC that are of value in
constraining the provenance and reactivity of OM (Griffith et al., 2010), other geochemical
characteristics of organic matter, including the elemental composition (e.g., C/N ratio)
abundance, stable isotopic ($^{13}$C, $^{15}$N) and molecular (biomarker) composition of organic matter,
as well as contextual properties such as sedimentation rate, mixed-layer depth, and redox
conditions (Aller and Blair, 2006; Arndt et al., 2013; Griffith et al., 2010) are needed to provide
a holistic depositional perspective. With on-going analytical advances that facilitate more
rapid and streamlined sediment analysis, it is anticipated that there will be substantial increases
in data availability and diversity, highlighting the urgent need to compile, organize and
harmonize existing datasets.

2. The MOSAIC database
In this study, we present MOSAIC (Modern Ocean Sediment Archive and Inventory of Carbon)
– a database designed to provide a window into the spatial variability in geochemical and
sedimentological characteristics of surficial ocean sediments on regional to global scales.
MOSAIC represents the starting point of an on-going endeavor to compile from data from prior
and on-going studies in order to build a comprehensive, continental margin-centric picture of
the distribution and characteristics of organic matter accumulating in modern ocean sediments.
The database infrastructure has been configured for facile incorporation of new data, for
expansion of included parameters, as well as for retrieval of data in an accessible and citable
format. MOSAIC is realized in an interactive web environment which allows users to visualize,
select and download data. This infrastructure is built using open-source (or optional open-
source) software (SI Table 1). The overarching goal is for MOSAIC to serve as a data platform





for the scientific community to explore the nature and causes of spatial patterns of
biogeochemical signatures in ocean sediments.

2.1. Database scope and content

*2.1.1. Spatial and depth coverage and georeferencing*
The focus of MOSAIC is on the coastal ocean (continental margins) with limited inclusion of
data from deep ocean settings.  Attention is also restricted to surficial sediments (nominally the
upper ~ 1m) that are most effectively sampled with shallow coring systems designed to recover
an intact sediment-water interface (e.g., hydraulically-damped multicorer, box corer). The
rationale is because of the focus on processes associated with deposition, early diagenesis, and
burial of organic matter, rather than on down-core investigations used for paleooceanographic
and paleoclimate reconstruction. Sediment depth profile data primarily used to examine
diagenetic profiles, and to constrain sedimentation rates, mixed layer depths, redox gradients,
as well as to determine carbon fluxes and inventories.

*2.1.2 Scope of data acquisition*
The data currently comprising the MOSAIC database was extracted from over two hundred
publications. No unpublished data is included in the on-line version, and the focus of the
database in this initial phase of implementation is on an initial suite of commonly measured
sediment parameters (e.g. sampling depth, carbon content and $\delta^{13}C$) that are available in high
abundance. A non-exhaustive list of the most important parameters cataloged in the MOSAIC
database can be found in Table 1. A more comprehensive list of parameters that are targeted
for inclusion in the near future can be found in the Supplemental Information (SI).

*2.1.3 Core parameters*
The database was established based on selected key parameters, with a particular emphasis on
the radiocarbon content of OC, as well as other basic properties that provide broader
geochemical and sedimentological context (Table 1).  The former include total organic carbon
(TOC) and total nitrogen (TN) content, organic carbon/total N ratios, and the stable carbon
isotopic composition ($\delta^{13}C$ and $^{14}C$ values) of OC. Sedimentological parameters are yet to be
implemented in the on-line version but will include parameters such as grain size, mineral



specific surface area, mixed layer depth, oxygen penetration depth, sedimentation rate, porosity
and dry bulk density.
2.2 MOSAIC Structure
The normalized relational database structure of the MOSAIC database was created using the
open-source MySQL software (MySQL Workbench Community for Ubuntu 18 version
6.3.10). The relational aspect of the database means that data (e.g., related to sample or
location-specifics) are stored in data tables which are connected (or related) by a unique
identifier. "Normalized" implies that in the structure of the database redundancies are
eliminated (e.g., a variable such as water depth occurs only once in the database, Codd, 1990).
A schematic of the detailed database structure can be found in SI Figure 2. The database
structure contains entries for key geochemical parameters pertaining to ocean sediment core
samples, including organic matter content, isotopic signature, and composition, as well as
texture and sedimentological parameters. Information can be collected for bulk samples as well
as for example size and density fractions. Furthermore, it is designed to enable additional
modules that can accommodate data related to other sample suites such as sinking particulate
matter from the ocean water column (e.g., time-series sediment traps), or riverine samples. It
includes is an exclusivity option which can be used to indicate if data is in the public domain
or not (e.g., pending publication of separate contributions).
Reporting conventions are detailed in the SI Table 2. Units as specified in the original papers
were used (listed in SI). Where possible $^{14}C$ information was collected as $\Delta^{14}C$, alternatively it
was collected as Fm and all $\Delta^{14}C$ values were converted to Fm (Stuiver and Polach, 1977).
Ongoing efforts are underway to further harmonize the data and convert all data to $\Delta^{14}C$ for
the next iteration for the MOSAIC database.
2.3 The MOSAIC Pipeline
There is a five-step pipeline for incorporation of data into MOSAIC. These are: (1) data
ingestion, (2) quality control, (3) transformation and structuring and (4) addition to a user-
friendly MySQL database interface, which is (5) available for users via a website (Figure 1).
This design enables users to query the collected data and augment and extend the existing
database using familiar spreadsheet software (Microsoft Excel®, LibreOffice). The associated
app allows any user to interactively select, visualize and query data without using database
(SQL) syntax (SI Figure 1).





*2.3.1 Data ingestion*
Input of data to the database is possible by filling in a pre-structured spreadsheet file with set
vocabularies. The user selects relevant parameter inputs from drop-down menus that streamline
data entry and assist in execution of subsequent SQL queries. Excel files were designed for
specific datasets, and within each Excel file there are three sub-tabs corresponding to groups
of the normalized MOSAIC SQL database (more details on database structure are provided in
the database). These tabs are (i) sample-related tab, (ii) geopoint-related tab (i.e., location), (iii)
author-related tab (i.e., paper). Certain variables pertaining to sample coordinates and depth
are required for data submission (i.e., latitude, longitude, water depth and sample core depth).
In this first version of MOSAIC, filled-in spreadsheet files with specified units and pre-defined
lists can be sent to mosaic@erdw.ethz.ch[1] for ingestion into the database.

*2.3.2 Data quality control*
Quality control of the input data is implemented via a python script tailored to the pre-defined
spreadsheet files. This script auto-checks the values of key parameters such as latitude,
longitude, carbon and nitrogen content, $^{13}$C, $^{14}$C, $CaCO_3$ content, $SiO_2$ content and sediment
texture-related parameters. The auto-check produces a log file with flags for unexpected values.
In turn, the flags point to the exact line containing possible out-of-bound values. For example,
for TOC (%), if values are negative, there will be a prompt "*cannot be negative, please check*",
when values are > 2 and <20 there is a prompt "*is quite high. Are you sure it is correct?*" and
lastly if values are > 20 there is the prompt "*value is high. Please check units*". Each flag is
accompanied by a line number to locate the possibly erroneous data. These flags then trigger a
manual quality check of the data by an expert in-house user.

*2.3.3 Data transformation and structuring*
The next step involves transforming data (using Python code) from Excel into csv files that are
compatible with the normalized relational database structure in SQL. This is done by (*i*) adding
unique identifiers to the data and (*ii*) transforming the data into appropriate csv files.
Importantly for the database structure, unique identifiers are created for each appropriate
database table (SI Figure 2). For example, for a specific location, an individual sediment core
may yield multiple samples (i.e., core sections corresponding to different depth intervals), with

---

[1] Data ingestion files MOSAIC_data_input_file.xlsx or MOSAIC_data_input_file.ods are available with this publication



multiple measurements (e.g., $^{13}$C, $^{14}$C and %TOC) performed on each sample (section). In this
example, the location is assigned a unique geopoint location identifier, the core receives a
unique identifier, and each sample (section) is given a unique identifier. These identifiers
resurface in each database table (e.g., on compositional parameters), resulting in the possibility
of multiple cores and multiple sample identifiers for a single geopoint. For the creation of
identifiers, the Python script finds a unique combination of coordinates (i.e., latitude and
longitude), assigns an identifier and eliminates duplicates. It repeats this for all primary keys
in the database.

*2.3.4 MySQL interface*
The Excel files designed for facile data ingestion are transformed in order to be compatible
with the normalized database using a Python script. This script executes this transformation by
auto-creating the compatible csv files, including the unique identifiers for the primary keys.
The script can be adapted to a dataset and is provided in the SI. The MOSAIC SQL database
allows for a direct upload of csv following data quality assessment, addition of identifiers and
creation of csv files. At present, a member of the ETH Biogeoscience group is allocated to
undertake this task upon receipt of files.

*2.3.5 MOSAIC Website: User access and citing of data*
The website (mosaic.ethz.ch) can be cited using the digital object identifier number (DOI)
https://doi.org/10.5168/mosaic019.1. In order to access data, users do not need to use SQL
syntax. Instead, users can select data of interest using drop-down menus or by selecting data
via a visual geographic interface. The selected data resulting from the query is shown in a table
and can be directly downloaded as a csv file (SI Figure 1). When querying data through the
MOSAIC website, the relational aspects of the database ensures that, for example, when a
certain location is selected, all data pertaining to this point appear in the table and are
downloaded. For users versed in SQL syntax, all accompanying data is available in SQL code,
which can be imported in both MySQL and PostgreSQL graphic user interface software. In
this format, all data can be queried in using SQL syntax.





## 3. Results and Discussion

### 3.1 Excerpts from the MOSAIC database

We provide examples of information extracted from MOSAIC (https://doi.org/10.5168/mosaic019.1, Van der Voort et al., 2019). The intention here is to illustrate broad-scale variability in OC properties rather that offer in-depth interpretations. The latter will be the focus of subsequent contributions.

We first explore the statistical distributions of geochemical properties (Figure 3). On a global scale, TOC contents of marine surface sediments (< 100 cm) are lognormally distributed around ~1 % (mean = 1.63%, median = 1.14%; n= 8688; Figure 3a), consistent with prior observations (Keil, 2017; Seiter et al., 2004, 2005). The distribution of stable carbon isotope ($\delta^{13}$C) values of OC shows two distinct populations (mean = -22.6‰, median = -22.18‰; n = 4297; Figure 3b), likely reflecting relative dominance of terrestrial C3 plant (~-27 ‰) and marine (~-22 ‰) sources (Burdige, 2005; Sackett and Thomson, 1963). Corresponding radiocarbon contents (expressed here as Fm values) exhibit a more unimodal distribution with an average Fm value of ~0.7 (Mean = 0.7, Median = 0.73, n = 709; Figure 3c), highlighting the significant proportions of pre-aged OC in globally distributed marine surficial sediments (Griffith et al 2010).

Carbon isotopic compositions of surface sediment OC exhibits substantial variability when plotted as a function of water depth (Figure 4). Radiocarbon contents are especially variable and generally lower in shallow (coastal) areas where TOC is also relatively low (Figure 4a). Coastal areas are both prone to supply of pre-aged OC from adjacent land masses (e.g. Tao et al., 2015; van der Voort et al., 2017), as well as ageing associated with sediment reworking by bottom currents (Bao et al., 2016). A similar pattern of variability is evident in $\delta^{13}$C values (Figure 4b) which exhibit a larger spread on continental shelves (~-13 to -30 ‰) and converge towards higher (more $^{13}$C-enriched) $\delta^{13}$C values (~- 22 ‰) in the deeper ocean. These trends reflect trajectories and modes carbon supply both from land and the ocean to the seafloor that govern OC sequestration and resulting sedimentary signatures (Bianchi et al., 2007; Burdige, 2005). Distinguishing between and quantifying the relative importance these factors is important for understanding consequences for carbon burial (Arndt et al., 2013; Bao et al., 2019; Bao et al., 2016), and requires ancillary geochemical and sedimentological (e.g., biomarker signatures, grain size distributions) information that will be incorporated into a future iteration of the MOSAIC database.



Broad-scale variability in OC characteristics of surface marine sediments also emerges
when properties are examined as a function of latitude (Figure 5). For example, despite
considerable scatter in stable carbon isotopic compositions, there is a general trend from higher
to lower $\delta^{13}C$ values with increasing latitude (Figure 5a). This could reflect latitudinal
variations in the carbon isotopic composition of marine phytoplankton (Goericke and Fry,
1994), and/or changes in the proportions and $\delta^{13}C$ values of terrestrial OC inputs (e.g., balance
of $C_3$ vs $C_4$ vegetation; Huang et al., 2000). Latitudinal trends in $^{14}C$ are less clear due to a
paucity of data with sufficient geographic coverage (Figure 5b), and serve to highlight ocean
regions and domains that are presently understudied with respect to this and other sediment
variables.

3.2 Scientific value of MOSAIC
The compilation of data and subsequent re-analyses holds the potential to yield novel insights
into the distribution and composition of OC accumulating in the contemporary marine
environment, shed light on underlying processes, and identify gaps in existing data sets. The
latter is particularly pertinent for $^{14}C$ data and ancillary measurements necessary to broadly
apply isotopically-enabled models of organic turnover and burial in sediments (e.g., Griffith
et al., 2010) and constrain geographic variability in the age distribution of sedimentary OC in
an analogous fashion to those of, for example, soil carbon (e.g. Shi et al., 2020). Filling such
gaps is also important given increasing interest in developing robust assessments of carbon
stocks in coastal marine sediments in the context of future greenhouse gas reporting protocols
(e.g. Avelar et al., 2017). Moreover, regional-scale data compilation of spatially
comprehensive geochemical and sedimentological information (Bao, et al., 2018; Bao et al.,
2016), coupled the application of novel numerical clustering methods (Van der Voort et al.,
2018) can facilitate refinement of criteria for delineating biogeochemically provinces
(Longhurst, 2007; Seiter et al., 2004), that reflect both source inputs and hydrodynamic
regimes, in order to improve carbon cycle budgets and models. Such examples highlight the
value of leveraging existing datasets, connecting various data sources and using other types of
analyses (modelling, statistics) in order to garner new insights into underlying processes.

3.3 MOSAIC in context.
MOSAIC complements other ongoing efforts to collect and organize a broad spectrum
geochemical and related data, such as the PANGAEA data repository (AWI and MARUM,



2020), as well as those with more targeted missions, such as the International Soil Radiocarbon
Database (ISRaD; Lawrence et al., 2020). It differs from these and other initiatives in its
targeted approach with a primary focus on (*i*) collating data pertinent to OC burial on
continental margins, (*ii*) upper sediment layers (nominally $< \sim 1m$) that encompass early
diagenetic processes and recent deposition, and (*iii*) radiocarbon information that bridges to
equivalent databases for other carbon cycle compartments.  The MOSAIC database has been
designed to be modular and adaptable to accommodate further developments and expansion of
its dimensionality, while retaining its overall carbon-centric focus. In particular, inclusion of
14C data on specific fractions separated, for example, according to sediment density
(Wakeham et al., 2009) or thermal lability (Rosenheim et al., 2008), or at the molecular level
(e.g. Druffel et al., 2010). In this context, it is anticipated that MOSAIC will serve as a key
research and teaching resource for biogeochemists focusing on contemporary biogeochemical
processes as well as seeking to interrogate sedimentary archives to develop records of past
oceanographic conditions.

4.  Data Availability
The data of the database can be accessed via mosaic.ethz.ch and the DOI is
https://doi.org/10.5168/mosaic019.1 (Van der Voort et al., 2019). Users who would like to add
data to the database can fill in the data in the Excel® templates that can be found in the SI of
this paper and send it to mosaic@erdw.ethz.ch.

5.  Conclusion and Outlook
In this paper, we introduce the motivation for development of a database (MOSAIC) focused
on OC accumulating in contemporary continental margin sediments. The structure of the
database and the associated web interface for data submission and retrieval is presented. The
supporting infrastructure was built with open-source software (SQL, R, Python, LibreCalc;
also provided with this contribution). Current data residing within MOSAIC derives from over
200 peer-reviewed papers, with the intention that this resource will further expand both
regarding data density and dimensionality, with a specific emphasis on radiocarbon as an
underdetermined yet crucial property for constraining carbon cycle processes. Construction of
parallel databases focused on riverine data and ocean sediment trap data are also under
development.



6.  Video Supplement
Accompanying this paper is a short instructional video (in SI) which explains users how to
download the data from MOSAIC (https://doi.org/10.5168/mosaic019.1, Van der Voort et al.,

372  2019).


7.  Author Contributions
Tim Eglinton led the conceptual development of the MOSAIC project. Tessa Sophia van der
Voort designed, structured and filled the SQL database and also created the associated
infrastructure in R, Python and Excel/LibreOffice. Thomas M. Blattmann and Daniel
Montluçon provided feedback on the database structure and website development and
contributed to discussion of the data. Mohammed Usman collected the MOSAIC data and
contributed to the data evaluation. Thomas Loeffler enabled the set-up of infrastructure and
contributed to the technical components of the paper. Maria Luisa Tavagna contributed to the
concept development. Nicolas Gruber contributed to the MOSAIC concept development and
project set-up. T.S. van der Voort prepared the manuscript with help of all co-authors.

8.  Competing interests
All co-authors declare that they have no competing interests regarding this manuscript.

9.  Acknowledgements
This project was funded by the ETH project (T. Eglinton and N. Gruber) "Elucidating processes
that govern carbon burial in the global ocean" (46 15-1). We thank Melissa Schwab for sharing
her insights in optimal R visualization. Many thanks also to Stephane Beaussier, who helped
to overcome numerous challenges in the development of this project. We thank Anastasiia
Ignatova for contributions to a prototype of MOSAIC. We thank Philip Pika for his insights
into sediment parameters.



10. Tables and Figures




*Table 1 Overview of key variables and their abundance in the MOSAIC database. An exhaustive list can be found in the SI.*

|  | Main variable | Unit | Number of datapoints | Required (Y/N) |
|---|---|---|---|---|
| **Geopoints** | Latitude | Degrees (°) | 8706 | Y |
|  | Longitude | Degrees (°) | 8706 | Y |
| **Samples Ocean** | Exclusivity Clause | Y/N | 8706 | Y |
|  | Water depth | m | 4297 | Y[2] |
|  | Sample core depth (average) | Centimeter (cm) | 7147 | Y |
|  | Sample name | VARCHAR | - | N |
|  | Total Organic Carbon (TOC) | Percentage (%) | 8688 | N |
|  | $\delta^{13}C$ | Permil (‰) | 4297 | N |
|  | Fm | fraction | 709 | N |
|  | C:N Ratio | Ratio | 504 | N |
|  | $SiO_2$ | Percentage (%) | 370 | N |
|  | $CaCO_3$ | Percentage (%) | 1668 | N |
| **Articles** | Article doi | VARCHAR | 235 | N |

[2] There are ongoing efforts to collect all water depth information, ancillary information will be attained using the GEBCO bathymetric grid (GEBCO, 2020).



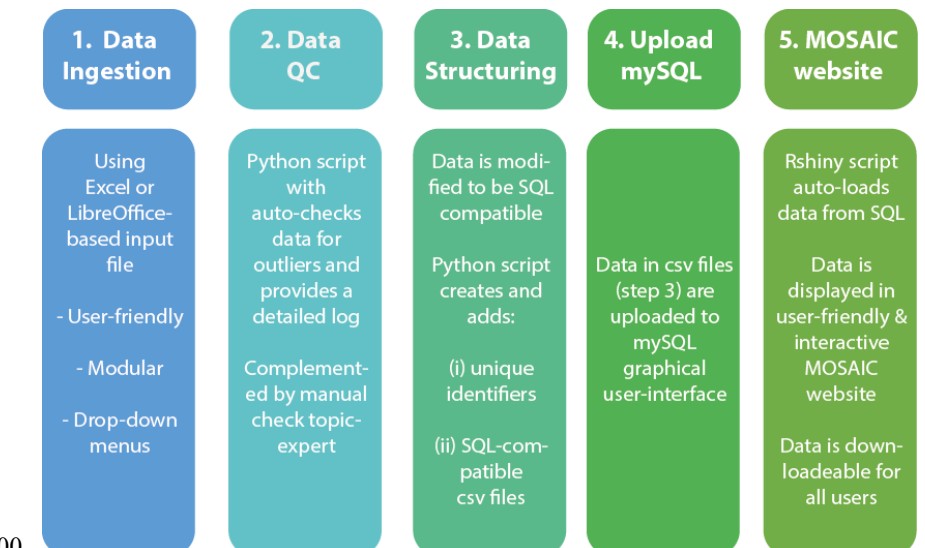


*Figure 1 Overview of the MOSAIC pipeline. Data ingestion (1) is done with excel-based input files. Then, (2) data quality control is achieved using is a python script which auto-checks the data for outliers and produces a subsequent log. Afterwards, (3) unique identifiers are added and the data is transformed into SQL-compatible format in Python. Subsequently, (4) data addition to the MOSAIC database occurs within the MySQL GUI, and finally (5), the data is auto-updated within the R environment and the Rshiny app is updated.*






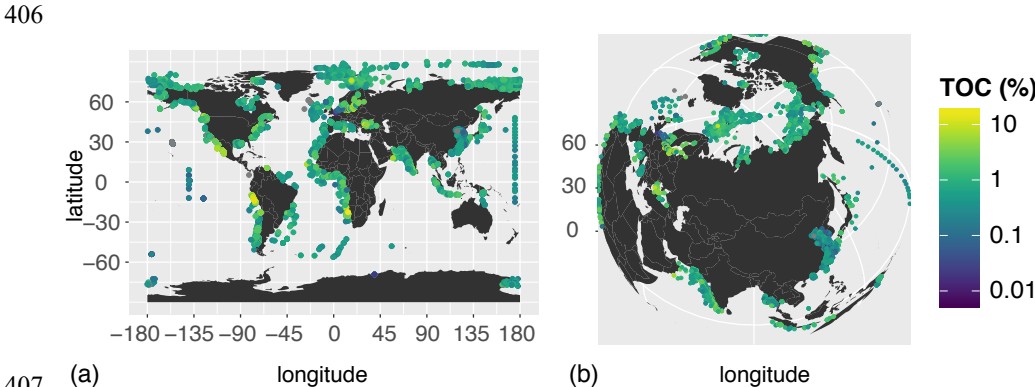

(a) longitude      (b) longitude

*Figure 2 distribution of all datapoints across the globe (a) from a standard projection and (b) from a polar-centric projection.*
*Colours indicate TOC content (%).*

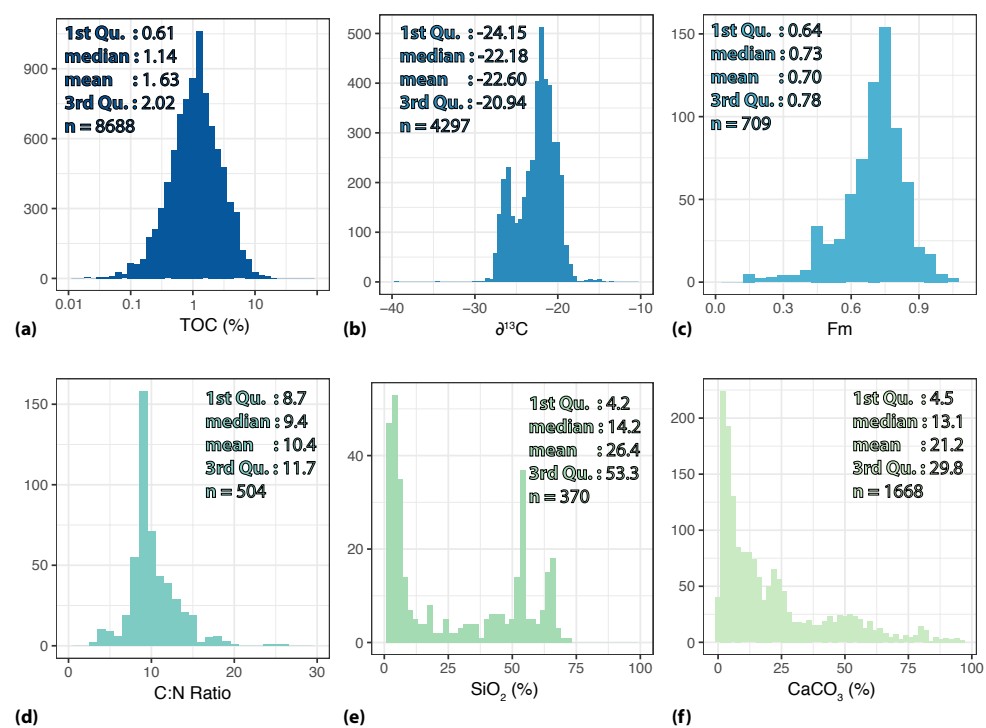

**(a)**    **(b)**    **(c)**

**(d)**    **(e)**    **(f)**


*Figure 3 Distribution of data for key sedimentary parameters included in MOSAIC: (a) TOC shows a log-normal distribution*
*which peaks at ~1.1 % and averages around 1.6 %, (b) δ¹³C values show two distinct peaks at ~-22 and ~27 permill. (c)*
*radiocarbon shows a strongly depleted signature with the fraction modern value averaging at ~0.7. The (d) C:N ratio global*
*average is ~ 10. The median (e) silicate (SiO₂) and (f) carbonate (CaCO₃) contents are ~14%, and ~ 13%, respectively*

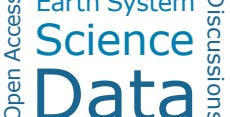


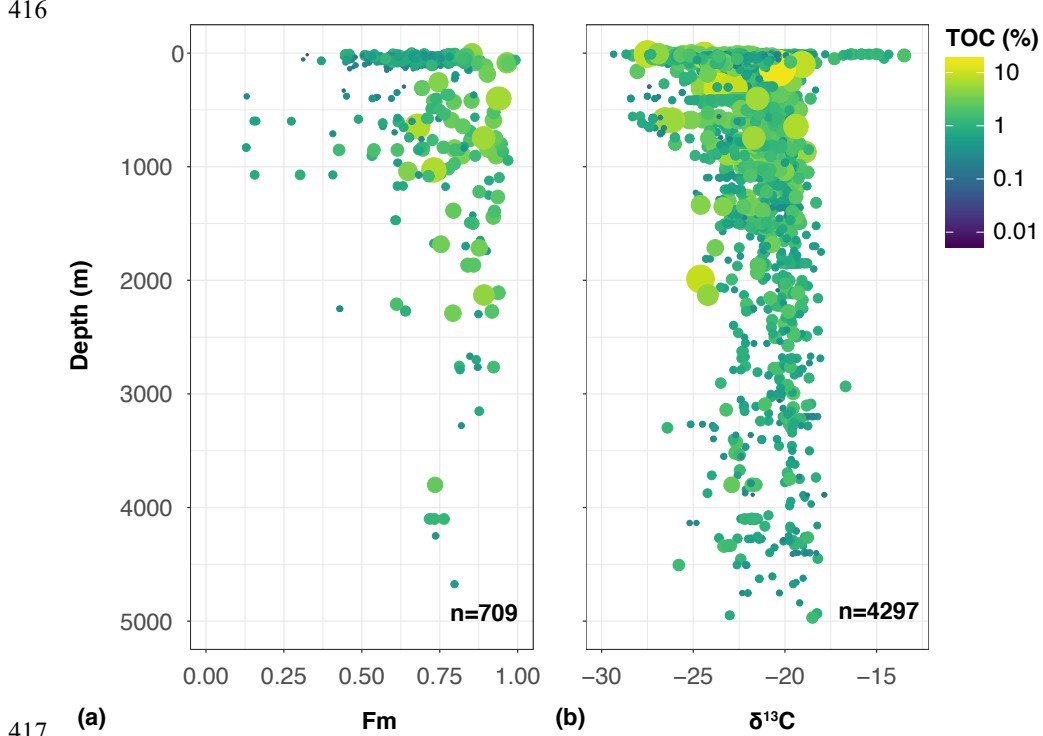

**(a)**           **Fm**        **(b)**         **δ¹³C**


*Figure 4 (a) Fraction modern versus depth, bubble size and colour indicate sample TOC content (%). On ocean shelves (shallow*
*depths) we observe generally low TOC values and depleted Fm values. Carbon in deeper oceans show a larger spread in ages*
*and TOC content. (b) δ¹³C modern versus depth, bubble size and colour indicate sample TOC content (%). On ocean shelves*
*(shallow depths) we observe a large spread in ∂¹³C values. Carbon in deeper oceans show a smaller spread and converge to*
*less depleted δ¹³C values.*

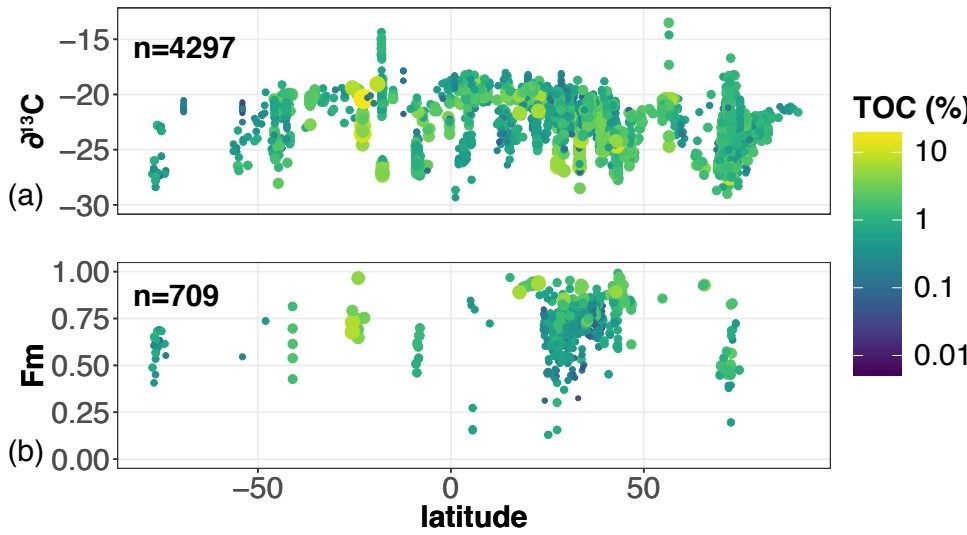


Figure 5 latitude (a) versus $\delta^{13}C$ and (b) Fraction Modern (Fm), colour indicated by TOC content (%). The $\delta^{13}C$ tends to be less

depleted in the low-latitudes. The Fm shows a sampling bias in the mid-range latitudes and also appears to be less depleted

in the lower latitudes.




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
