# Peer review of "MOSAIC (Modern Ocean Sediment Archive and Inventory of Carbon)"

_Earth System Science Data, 2020_

## Referee Comment (RC1) · Anonymous Referee #1 · 8 Dec 2020

I enjoyed reading the manuscript by T.S. van der Voort et al. entitled "MOSAIC (Modern Sediment Archive and Inventory of Carbon): A (radio)carbon-centric database for seafloor surficial sediments". The need for a surficial sediment database for organic carbon and radiocarbon is well justified in the text. I would like to see the MOSAIC database established, and as an indication of my support and approval, I am likely to contribute most of my radiocarbon and organic carbon data to such a program. I like the idea of using open-source software and making this focused database convenient to the scientific community and user friendly.

[Figure]

The only concern I have about MOSAIC, as it strives to become accepted as a global database, is that I wasn't certain as to which 200 papers were used to establish the initial database. The authors make generalizations regarding C-13 and C-14 data in the discussion section, but the rigor of these generalizations depends on which 200 papers were used to establish the data base. Were these primarily papers written by Tim Eglinton's research group or was a broader approach used in the selection of the organic carbon and radiocarbon data? There is reasonable global coverage of continental margin sedimentation in the MOSAIC data, but there are some obvious holes in the database, such as the continental margin sediments surrounding the Antarctic Peninsula (where there has been substantial radiocarbon data published in the past several years).

The manuscript does describe QA/QC concerns of the radiocarbon and total organic carbon data, but these are primarily from a statistical perspective. Very little is mentioned in the manuscript about analytical concerns, blank issues, and potential contamination during sampling. The database currently lists radiocarbon data using a Fraction Modern (Fm) nomenclature, but mentions that Delta14C nomenclature will ultimately be used for the database. If so, I would recommend that the authors include a "Date of Collection" data box in their submission data and website display, so that users can easily go back and forth between Fm nomenclature and Delta14C nomenclature. In fact, I think it would be useful to list the radiocarbon data using both the Fm and Delta 14C formats. In addition, I also would recommend that the authors consider adding to the data input table the type of coring device used to collect the marine sediments. There is a big difference between the quality of surficial sediment collected by a multicorer or megacorer as compared to a kasten corer or piston corer. Such information would be useful to a researcher comparing organic carbon or radiocarbon abundances over a basin or region.

I think that the authors make the case that radiocarbon data are the most needed information for continental margin databases. That being said, of the total 8706 data

entered into MOSAIC, there are only 709 radiocarbon measurements (as compared to 8688 analyses of Total Organic Carbon). Thus, although radiocarbon may be the primary emphasis of the MOSAIC database, it represents less than 10% of the data entered into the system. The MOSAIC database also lists the Calcium Carbonate content and the Silicate ($SiO_2$) content of the sediments. The text does not reference how these measurements were made or even if the silicate abundances includes biogenic silica with the lithogenic silica content.

Minor Suggestions and Concerns:

1. The manuscript could have been proofread more thoroughly prior to submission. For example: -On lines 159-161 the words don't comprise a complete sentence. -On line 193 add commas on either side of "for example". -On line 276 the text reads "rather that" and it should be "rather than". -On line 289 "exhibits" should be "exhibit". -On line 327 change "couple" to "couple with". -On line 336 add "of" before "geochemical". -On line 345 change "14C" to "14ˆC". -On line 363 change "derives" to "was derived". -On line 370 change "explain users" to "explain to users". -In Fig. 5 the partial derivative sign is used instead of the small Greek symbol delta. The Greek symbol is used correctly in Fig. 4, but the partial derivative symbol needs to be changed to a lower case delta symbol in Fig. 5.

2. On lines 128 and 179, the authors should consider not only listing the "mixed-layer depth", but also include "bioturbation intensity" as a parameter for characterizing the nature of surficial sediments.

3. On line 293 the text states: "ageing associated with sediment reworking by bottom currents". The authors should mention bioturbation as well as physical sediment reworking. It is much more likely that continental shelf and continental margin sediments are mixed by bioturbation than by physical reworking.

4. On lines 318-322 the text reads: "The latter is particularly pertinent for 14C data and ancillary measurements necessary to broadly apply isotopically-enabled models

of organic turnover and burial in sediments (e.g., Griffith et al., 2010) and constrain geographic variability in the age distribution of sedimentary OC . . .". I suggest that the authors consider adding the following reference after the Griffith et al., 2010 citation: Isla and DeMaster, 2018 (GCA, v. 242, 34-50; entitled "Labile organic carbon dynamics in continental shelf sediments after the recent collapse of the Larsen ice shelves off the eastern Antarctic Peninsula: A radiochemical approach"). This paper is a recent example of "isotopically-enabled models of organic turnover".

5. Why do the authors use the word "seafloor" in the title instead "marine". Using "seafloor" and "sediment" so close to each other seems redundant to me.

In summary, I support publication of the MOSAIC ESSD article after minor concerns, mentioned in the review above, have been addressed by the authors. I encourage the authors to continue their efforts to develop and create these new databases that enable scientists easier/facilitated access to organic carbon and radiocarbon data published in the marine science literature.

-

---

## Referee Comment (RC2) · Anonymous Referee #2 · 5 Jan 2021

In general, I welcome the proposed database and can see its value and utility. However, I do have several points to raise to the authors that should be addressed before publication:

1. The narrative in the Introduction forms a case for support for the need and uniqueness of the database on the one hand, whilst on the other slips into scientific argument of what could/should be done with the data. Both articulations are reasonable, but confuse the reader somewhat. I suggest toning down the suggestions on what can be done with the data. Overall, the paragraphs starting line 82 and line 120 seem largely

redundant. Similarly, i was surprised not to see reference to recent reviews and opinion pieces about sediment carbon (e.g. Snelgrove et al. 2018, TRENDS IN ECOLOGY & EVOLUTION, 33, 96-105; Middelburg, 2018 BIOGEOSCIENCES 15, 413-427) to re-inforce what we know and what we don't know. these, and other similar summaries should be incorporated into the text.

2. MOSAIC - minor point, but this acronym is a little unfortunate as it matches the MO-SAIC expedition in the Arctic (https://mosaic-expedition.org/ ), a significant programme that will have a long legacy in the literature. I suggest altering the acronym to avoid this overlap, and suggest the authors consider using a title rather than an acronym that incorporates the description of the exactly what is in the database.

3. Line 146 (and then Line 170)- I see the intention of the database, but how often will it be updated and what data quality controls are in place? re line 170, how with the new information gel with the older data, and will efforts be made to back fill the missing data?

4. Paragraph starting Line 164 - A very important aspect of any database that has ex-tracted information from the literature is that the search terms and process of selection criteria needs to be repeatable and absolutely clear. This is of fundamental importance and needs to be explicitly stated in the this section with supporting information in the supplementary material. How were the 200 papers found, selected and checked for data? What search engines and search terms (including any refinements) were used, and how were quality controls implemented? How many papers did the initial search yield, and how was the final subset arrived at? When was the database accessed? Does this database contain data from other databases? What downstream processing of the data, or meta-data, was necessary? e.g. were units converted, how was lat and long derived/converted to the same projection, how was a position assigned to biogeographical zones etc? All steps need to be explained. This is an essential area that needs to be articulated in detail to ensure the authority of the data. The authors need to convince the reader that these data are the ones to use. This is probably the

most important aspects of my commentary that needs addressing fully. Section 2.1.2 needs significant amendments with a focus on attention to detail.

5. Line 177-180 - this is admirable and will be beneficial, but at present does not exist. This aspiration should be omitted from the current description. Instead, the authors should add in the Data Accessibility section that updates will take place (how often? when?) and how to access the latest version of the database. I assume that each iteration will have a documented history and version number thats traceable? If not, this needs to be implemented from the outset.

6. Line 186 - can each individual datapoint be traced back to the individual source (paper)? It will be important that users of the data can look at the context of each datapoint by going back to the original source if necessary. in other words, is there a unique identifier that matches the data value to the specific paper from which it was extracted? This is essential and needs to be included if not already done so.

7. Line 221 - how are submitted data quality checked? make this clear here.

8. Line 228 - how exactly are unexpected values determined? How is this reconciled with unexpected, or outlier variables, that are nevertheless real? Need to reassure the reader that the data is not being sanitised to some pre-determined criteria or parameters.

9. Data quality control - this section needs expanding, as stated earlier, to include quality controls at the point of data collection. The current section only lists quality control post collection. In addition, this section would benefit from some explanation/justification of the detail, supported by citations where necessary/appropriate.

10. Section 2.3.5 - it would be beneficial for the supplementary material to include an "idiots guide" for how to complete a search and extract the data for a simple and more complex query example. For example, what are the step through processes to extract a global dataset versus just one region, or whatever is likely to be a common query.

[Figure]

This should be made readable and accessible to users that have never used SQL or programming, or that have little or no experience of extracting data. The video is a useful addition in this regard, but a manual type addition to the supplementary material would be helpful.

11. Section 3.1 - much of this section is unnecessary and not particularly helpful. the description of the distribution of data is only relevant to the database as it now stands, but as highlighted in the papers, the database will be updated. hence, such statements will be misleading at the point of the first update. Instead, purely descriptive statistics that relate to the database structure (i.e. not interpretative information) should be presented, such as the number of observations for each variable, categorised by region, water depth and other column headings in the database. Presently, it is hard for the reader to understand what the database contains without entering the database itself. As made above (point #1), this section morphs from being a database description to a paper thats interpreting the data. In my opinion, as interesting as the summaries are, the latter has no place here. If the authors wish to interpret the data, they should write a separate contribution and publish elsewhere.

12. Section 3.2 - this can be condensed significantly, many of these points have been made in the Abstract and Introduction. The text would also benefit from reaching out to other fields, perhaps offering other areas that these data may be relevant to that have not received attention previously.

13. Section 3.3 - this section is quite weak and not very compelling. It is not entirely clear whether (i) the data contained in this database is a subset of the other databases mentioned, (ii) how these data differ from other inventories and what the pros and cons of these data are in relation to specific areas of research (maybe include reference to other databases that may form good companions to these data), (iii) and why a user should opt for using these data? Some aspects of these matters are listed, but only in very general terms that lack specifics. Much more explicit arguments need to be made here.

14. Section 4 - add a sentence that states what version of the database this paper is referring to/describing, and how often users can expect updates to the database (e.g. periodically, annually?). I suggest it will also be advantageous to state how errors can be reported.

15. Section 5 - this section is repetitive of the sections above and does not add anything new. This section needs revising to pick up from where the Introduction left off.

16. Table 1- the database contains 8706 entries with latitude and longitude, but only about half of these have a water depth associated with them - could those that do not have a water depth be estimated using, for example, Google earth based on the lat and long co-ordinates? I note the comment re GEBCO, but the same comment made earlier about the state of the database at the point of publication versus aspirations stands.

Overall, I am supportive of the communication, but as the manuscript now stands it does not include sufficient detail about how the data were derived and forms an in-compatible mix of existing versus aspirational database properties. I would see both of these as moderate revisions.

---

## Referee Comment (RC3) · Tessa Sophia van der Voort et al. · 7 Jan 2021

The MOSAIC database will be an extremely valuable tool for research in marine carbon storage. It is easily used and expandable. I am not an expert in marine sediments but I was glad to see the authors are thinking about adding on 14C of sediment fraction from density separations or thermal decomposition which can contribute important information on carbon source which of course would require fields for the density or temperatures used for each 14C measurement. I found the examples useful though obviously more data is needed. I definitely support publication of the MOSAIC article

in ESSD after minor revisions.

I have only a few very minor comments/corrections: Line 73: Why reference Reimer et al. 2009 when the 14C archives have been updated in Reimer et al 2020 (doi: 10.1017/RDC.2020.41)? Line 159-161: 'Sediment depth profile data primarily used to examine diagenetic profiles, and to constrain sedimentation rates, mixed layer depths, redox gradients, as well as to determine carbon fluxes and inventories'. This is an incomplete sentence. Line 176-177: 'and the stable carbon isotopic composition (d13C and 14C values) of OC'. Delete 'stable' since 14C is not stable. Line 199: Presumably $\Delta$14C is not age corrected but, in any case, the date of the 14C measurement would be needed to convert to Fm. If it is age corrected then the year of collection is also needed. Line 327: 'coupled the application. . .' Presumably this should be 'coupled with' Line 345: '14C' should have a superscript 14C.

---

## Author Comment (AC1) · 1 Mar 2021

Dear Editor,

Thank you very much for relaying these reviews to us. We are very encouraged that the reviewers are so positive, and they see the value of this new database. We have incorporated all revisions, and we greatly appreciate the time the reviewers have taken to provide detailed and constructive comments. The major improvement, suggested by both reviewers #1 and #2, was to provide additional details with respect to the data collection process and data conversions (e.g., of the coordinate systems). We agree, and we have now both included a short summary in the main text and have provided more details in a dedicated section in the

Supplemental Information. Please see the point-by-point replies below for details.

Again, we greatly appreciate the detailed feedback, and we hope that the revised manuscript, and the database described therein, will provide a foundation for further research in marine carbon and in the wider earth sciences.

Many thanks in advance for considering this revised manuscript,

On behalf of all co-authors,

Tessa van der Voort

Point-by-point replies referee #1

I enjoyed reading the manuscript by T.S. van der Voort et al. entitled "MOSAIC (Modern

Sediment Archive and Inventory of Carbon): A (radio)carbon-centric database for seafloor surficial sediments". The need for a surficial sediment database for organic carbon and radiocarbon is well justified in the text. I would like to see the MOSAIC database established, and as an indication of my support and approval, I am likely to contribute most of my radiocarbon and organic carbon data to such a program. I like the idea of using open-source software and making this focused database convenient to the scientific community and user friendly.

Thank you for this! We look forward to incorporating your data. That's how we can help
MOSAIC grow!

The only concern I have about MOSAIC, as it strives to become accepted as a global database,
is that I wasn't certain as to which 200 papers were used to establish the initial database. The
authors make generalizations regarding C-13 and C-14 data in the discussion section, but the
rigor of these generalizations depends on which 200 papers were used to establish the data
base. Were these primarily papers written by Tim Eglinton's research group or was a broader
approach used in the selection of the organic carbon and radiocarbon data?

Thank you for this suggestion. We have included a short description of how the papers were
selected in the main text, and a detailed description in the SI. A broader approach was used to
select the organic carbon and radiocarbon data, building upon an initial synthesis effort
(Griffith et al., 2010) where papers (not only those from the Eglinton research group)
containing TOC and $^{14}$C data from several margins was used as a starting point. This was then
augmented by a subset of papers containing similar information (i.e., sediment TOC and $^{14}$C
data) selected by the senior author (Eglinton), which were used to familiarize the researcher
involved (Usman) in extraction and assessment of relevant data. Then, additional papers were
sought out using google scholar, using search terms such as "TOC in surficial sediments",
"organic carbon in surficial sediments" and "$^{14}$C/Radiocarbon in surficial sediments" and
relevant data were ingested into MOSAIC. Where possible, references in found papers were
followed up on to access more or original datasets. This resulted in over 200 papers.

There is reasonable global coverage of continental margin sedimentation in the MOSAIC data,
but there are some obvious holes in the database, such as the continental margin sediments
surrounding the Antarctic Peninsula (where there has been substantial radiocarbon data
published in the past several years).

We are aware that our data search and ingestion process has far has not been exhaustive, due
to the limitation of personnel dedicated to this activity. The database is built to be dynamic and
can continually absorb more datasets, and it is intended to growth both through further
combining of the literature for additional data, through on-going acquisition of new data, and
as scientists become aware of MOSAIC and contribute their own data. It has been noted for the future updates. MOSAIC makes preliminary generalizations based on the 200 papers, but this only represents a fraction of the available literature, particularly given that our search terms may have missed numerous contributions. At this point, however, with this data collection and the developed digital infrastructure, we felt it timely to prepare an initial publication. A newly funded post-doc position in the Eglinton Group (starting March 2021) will be dedicated to the

MOSAIC database.

The manuscript does describe QA/QC concerns of the radiocarbon and total organic carbon data, but these are primarily from a statistical perspective. Very little is mentioned in the manuscript about analytical concerns, blank issues, and potential contamination during sampling.

Thank you for these comments. Indeed, there is a lot of focus on the statistical and automated

QA/QC. When the data was collected form the papers, care was taken to take from trusted, peer-reviewed sources. However, we also have to trust the quality of the researchers, labs and peer-review with respect to their data processing and reporting. Whenever it was available, we included reported uncertainties (error values). Due to space limitations these are not included on the website-based version of MOSAIC. However, in the SQL-based database, which is also included, this data can be accessed. We included an introductory guide in the SI on how to use specific database queries. We have added an example that also extracts all error values. For the most common parameters of course, the MOSAIC website is designed to provide a user- friendly, intuitive interface.

The database currently lists radiocarbon data using a Fraction Modern (Fm) nomenclature, but mentions that Delta14C nomenclature will ultimately be used for the database. If so, I would recommend that the authors include a "Date of Collection" data box in their submission data and website display, so that users can easily go back and forth between Fm nomenclature and

Delta14C nomenclature. In fact, I think it would be useful to list the radiocarbon data using both the Fm and Delta 14C formats.

Thank you for this comment. Regarding, Fm versus Delta$^{14}$C, whenever it was possible, the sampling year was known or collected, we converted Fm to Delta$^{14}$C and vice versa. For the future data ingestion (also from our fellow researchers), we have a field in the submission excel-sheet that includes the sampling year, so this conversion can be done. On the MOSAIC

website (mosaic.ethz.ch), both Fm to Delta$^{14}$C data are directly available.

In addition, I also would recommend that the authors consider adding to the data input table the type of coring device used to collect the marine sediments. There is a big difference between the quality of surficial sediment collected by a multicorer or megacorer as compared to a kasten corer or piston corer. Such information would be useful to a researcher comparing organic carbon or radiocarbon abundances over a basin or region.

Thank you for this comment. Whenever available, we collected data on the coring device, and this is also included in the database, but for simplicity this information is not included in the web-based interface. We provided a specific example on how to retrieve information on the corer type in the available introductory guide on how to use specific SQL queries.

I think that the authors make the case that radiocarbon data are the most needed information for continental margin databases. That being said, of the total 8706 data entered into MOSAIC, there are only 709 radiocarbon measurements (as compared to 8688 analyses of Total Organic

Carbon). Thus, although radiocarbon may be the primary emphasis of the MOSAIC database, it represents less than 10% of the data entered into the system.

Thank you for this comment. Due to the high monetary and labor costs, $^{14}$C measurements remain much rarer than TOC measurements, and the proportion of $^{14}$C to TOC data in MOSAIC

reflects that found in most papers. Most papers contain around a handful or a dozen $^{14}$C

datapoints, so even to attain this value a considerable effort was needed, and we intend to keep adding to it. However, this situation is changing rapidly as the number of accelerator mass spectrometry systems installed around the world has dramatically increased within the past decade. A key priority in the future development of MOSAIC is to incorporate newly reported and recently acquired $^{14}$C data, and this will be a primary focus of a newly funded post-doc position (starting March 2021).

The MOSAIC database also lists the Calcium Carbonate content and the Silicate (SiO2) content of the sediments. The text does not reference how these measurements were made or even if the silicate abundances includes biogenic silica with the lithogenic silica content.

The focus of data collection activities thus far have been on the abundance and characteristics
of organic matter, – however, the relevance of inorganic components with respect to, for
example, biogeochemical fluxes and organo-mineral associations is also recognized. Thus far,
no distinction has been made between biogenic (opal) or lithogenic silicate sources, but this
may be further defined in subsequent iterations of MOSAIC. In the meantime, if researchers
want to further explore such parameters, the DOI is provided for each datapoint, and they can
easily attain more details about the data with one click.
Minor Suggestions Reviewer #1
The manuscript could have been proofread more thoroughly prior to submission. For example:
-On lines 159-161 the words don't comprise a complete sentence.
Corrected, thank you
-On line 193 add commas on either side of "for example".
Corrected, thank you
-On line 276 the text reads "rather that" and it should be "rather than".
Corrected, thank you
-On line 289 "exhibits" should be "exhibit". Corrected, thank you
-On line 327 change "couple" to "couple with". Corrected, thank you -On line 336 add "of"
before "geochemical". Corrected, thank you -On line 345 change "14C" to "14^C". Corrected,
thank you -On line 363 change "derives" to "was derived". Corrected, thank you -On line 370
change "explain users" to "explain to users". Corrected, thank you -In Fig. 5 the partial
derivative sign is used instead of the small Greek symbol delta. The Greek symbol is used
correctly in Fig. 4, but the partial derivative symbol needs to be changed to a lower case delta
symbol in Fig. 5. Corrected, thank you
On lines 128 and 179, the authors should consider not only listing the "mixed-layer depth", but
also include "bioturbation intensity" as a parameter for characterizing the nature of surficial
sediments
We have included this, thank you.

. On line 293 the text states: "ageing associated with sediment reworking by bottom currents".
The authors should mention bioturbation as well as physical sediment reworking. It is much
more likely that continental shelf and continental margin sediments are mixed by bioturbation
than by physical reworking.
Thank you for this comment. In the paper cited here, the focus was on physical reworking of
sediments via lateral redistribution, however the reviewer is absolutely right that bioturbation
is an extremely important consideration. We have highlighted the importance of bioturbation
in line 128. Incorporation and parameters and data related to bioturbation (e.g., sediment mixed
layer depth, oxygen penetration depth) will be a focus of the next iteration of MOSAIC.
On lines 318-322 the text reads: "The latter is particularly pertinent for 14C data and ancillary
measurements necessary to broadly apply isotopically-enabled models of organic turnover and
burial in sediments (e.g., Griffith et al., 2010) and constrain geographic variability in the age
distribution of sedimentary OC . . .". I suggest that the authors consider adding the following
reference after the Griffith et al., 2010 citation:
Isla and DeMaster, 2018 (GCA, v. 242, 34-50; entitled "Labile organic carbon dynamics in
continental shelf sediments after the recent collapse of the Larsen ice shelves off the eastern
Antarctic Peninsula: A radiochemical approach"). This paper is a recent example of
"isotopically-enabled models of organic turnover".
Thank you, we concur, and have included a reference to this informative paper.
Why do the authors use the word "seafloor" in the title instead "marine". Using "seafloor" and
"sediment" so close to each other seems redundant to me.
Thank you for the comment. In the title we want to make clear that the focus of this database
is on surficial marine sediments (i.e., not the longer cores such as those acquired IOPD cruises
for paleoclimatic studies). Thus, we opt to use "seafloor" as we believe it best implies we are
discussing surficial ocean bottom sediments.

In summary, I support publication of the MOSAIC ESSD article after minor concerns,
mentioned in the review above, have been addressed by the authors. I encourage the authors to
continue their efforts to develop and create these new databases that enable scientists
easier/facilitated access to organic carbon and radiocarbon data published in the marine science
literature.

Thank you, we very much appreciate the detailed and constructive comments and the time
you've taken to provide them. We also appreciate the sentiment that this contribution will
further scientific research on ocean sediments. We have processed the comments and look
forward to sharing this work with the scientific community.

Anonymous Referee #2

Major comments:

In general, I welcome the proposed database and can see its value and utility. However, I do have several points to raise to the authors that should be addressed before publication:

Thank you for this comment. We've addressed the points that have been raised, and you are very grateful for the time and effort you have put into this review.

The narrative in the Introduction forms a case for support for the need and uniqueness of the database on the one hand, whilst on the other slips into scientific argument of what could/should be done with the data. Both articulations are reasonable, but confuse the reader somewhat. I suggest toning down the suggestions on what can be done with the data. Overall, the paragraphs starting line 82 and line 120 seem largely redundant. Similarly, i was surprised not to see reference to recent reviews and opinion pieces about sediment carbon (e.g. Snelgrove et al. 2018, TRENDS IN ECOLOGY & EVOLUTION, 33, 96-105; Middelburg, 2018

BIOGEOSCIENCES 15, 413-427) to reinforce what we know and what we don't know. these, and other similar summaries should be incorporated into the text.

Thank you for these comments, we have endeavored to incorporate your suggestions.

Indeed, we agree with your suggestion that the focus of this paper lies on presenting the

MOSAIC dataset and digital infrastructure. Citations for the mentioned opinion and summary papers (Snelgrove et al., 2018 and Middelburg 2018), which have now been incorporated in order to underline the broader utility of MOSAIC for the scientific community, e.g., by improving the robustness of sedimentary organic carbon turnover estimates and the understanding of organic matter processing in seafloor sediments. We have now also included additional summary papers, such as those by Arndt et al., (2013) and Bianchi (2011) that highlight the need for the type of information residing in MOSAIC.

Presently the paper is structure that it provides only a cursory glance at the available data by way of illustration of the sorts of information that can be retrieved. We have visualized the data, but it is not intended to be a rigorous assessment or provide in-depth interpretation (e.g.

quantifying carbon stocks or using machine learning algorithms to extract spatial patterns as e.g. we have done in respectively Avelar et al., (2017) and van der Voort et al., (2018)). We
now emphasize this point, and also point out that we have modelled the structure of this
manuscript to follow others in this journal that announce a database and provide examples of
data content.

2. MOSAIC - minor point, but this acronym is a little unfortunate as it matches the MOSAIC
expedition in the Arctic (https://mosaic-expedition.org/ ), a significant programme that will
have a long legacy in the literature. I suggest altering the acronym to avoid this overlap, and
suggest the authors consider using a title rather than an acronym that incorporates the
description of the exactly what is in the database.

Indeed, MOSAIC is an acronym that occurs in other settings, and indeed also for the Arctic
expedition. The latter is a field program, while our website (mosaic.ethz.ch), clearly
immediately refers to being a database. We believe scientists will be able to readily make this
distinction. Furthermore, the capitalization of both abbreviations is different, where the
database is all caps (MOSAIC), the expedition has a lower-case "I" (MOSAiC). We believe
MOSAIC is an apt name, because we investigate spatial mosaics in geochemical and
sedimentological properties on the ocean floor.

3. Line 146 (and then Line 170)- I see the intention of the database, but how often will it be
updated and what data quality controls are in place?

Thank you for this comment. We aim to announce a quasi-yearly update, with the most up-to-
date version mentioned on the website. Starting March $1^{st}$, a dedicated post-doc will be fully
devoted to this project, with a focus on further ingestion of data (esp. $^{14}C$) and expansion of
parameters. Regarding the quality controls, (as mentioned in section 2.3.2), we have an initial
auto-check written in Python which checks data and flags unusual data (e.g. TOC values that
are <0). We have now added a detailed description of all automated checks in the SI. After the
automated check, a member of the ETH MOSAIC Team will manually check the flagged data.
This ETH MOSAIC team member will also perform an additional manual check to see if all
the data was read in correctly.

re line 170, how with the new information gel with the older data, and will efforts be made to
back fill the missing data?
MOSAIC has been explicitly designed to be a dynamic database. Data can be added and
ingested easily, as described below. As part of these efforts, there will be targeted efforts to
"back-fill" missing data, as we continue to uncover previously published work. Indeed, this is
one of the defining attributes of MOSAIC – that it has a specific objective to collate and
organize data germane to the overall theme of organic matter accumulation on continental
margins, instead of serving as a passive repository for data. We are also open to incorporating
new variables for future versions (e.g., those relevant to seafloor ecology) if they are brought
forward by the community. Thanks to the adaptable SQL framework, this would involve just a
few lines of new code.
New data can be ingested in the provided spreadsheets which have built-in vocabularies (e.g.,
for corer types or ocean names). Then, the data in spreadsheets (Microsoft or LibreOffice) will
be converted to be SQL-ingestible. This is done by using Python scripts that automatically add
unique identifiers to the data and convert the Microsoft Excel or LibreOffice files to csv files
which can be ingested in the mySQL environment.
We aim to continually expand the MOSAIC dataset. For example, a key next goal is to develop
carbon inventories of sediments according to the Economic Exclusive Zones (EEZs), and to
identify regions where data is particularly sparse. The continuous addition of data (new and
old) to MOSAIC will enhance the value for the scientific community.
More technical details on Quality Control are section 2.3.3.

4. Paragraph starting Line 164 - A very important aspect of any database that has extracted
information from the literature is that the search terms and process of selection criteria needs
to be repeatable and absolutely clear. This is of fundamental importance and needs to be
explicitly stated in the this section with supporting information in the supplementary material.
How were the 200 papers found, selected and checked for data? What search engines and
search terms (including any refinements) were used, and how were quality controls
implemented? How many papers did the initial search yield, and how was the final subset
arrived at? When was the database accessed? Does this database contain data from other
databases? What downstream processing of the data, or meta-data, was necessary? e.g. were
units converted, how was lat and long derived/converted to the same projection, how was a
position assigned to biogeographical zones etc? All steps need to be explained. This is an
essential area that needs to be articulated in detail to ensure the authority of the data. The
authors need to convince the reader that these data are the ones to use. This is probably the
most important aspects of my commentary that needs addressing fully. Section 2.1.2 needs
significant amendments with a focus on attention to detail.

Thank you for this comment. We fully concur with this point and have revised and expanded
the text accordingly.
We have included a brief summary answering the issues you raised in the main text and added
a highly detailed section in the supplemental information.
To answer your question directly here:
**Q1A: How were the 200 papers found**

The current MOSAIC dataset was initiated by manual mining of an initial subset of
peer-reviewed oceanographic papers that contained substantial $TO^{14}C$ datasets (e.g.,
Griffith et al., 2010) from different continental margin systems. This initial dataset was
collected by an experienced oceanographer, this papers' senior author (Eglinton) This
enabled the collecting researcher (Usman) to be trained in the process of data evaluation
and handling.

MOSAIC was further expanded by extracting data from a broader suite peer-reviewed
papers which were found using the search engine Google Scholar, with search terms
including "organic carbon in surficial/surface sediments", "TOC in surficial/surface

| 333 | sediments" and "radiocarbon/$^{14}$C in surficial/surface sediments".". When appropriate |
| 334 | papers were found, references were followed up on to find similar contributions in the |
| 335 | region. This yielded several hundred of papers. |

**Q1B selected?**

| 337 | From the several hundreds of papers, only papers that contained the required parameters |
| 338 | for data were retained (i.e., lat. and long. for each TOC or $^{14}$C data point). Furthermore, |
| 339 | the papers which focused on sediment dissolved organic carbon or inorganic carbon |
| 340 | were excluded given the focus on the solid phase and organic phase, and a priority on |
| 341 | surficial sediment data, captured by corers that best preserve the sediment-water |
| 342 | interface (i.e., multicorer or box corer.). Other corers are not strongly represented. |
| 343 | Furthermore, papers were selected for whom the data was available in tabulated for, i.e. |
| 344 | not exclusively in graphical form in order to ensure the quality of extracted data (see |
| 345 | next paragraph for more details). |

**Q1C and checked for data?**

| 349 | As mentioned, for the older papers, the researcher extracted the data manually, point- |
| 350 | by-point from tables or exceptionally from graphs in papers in pdf format. While this |
| 351 | process is very laborious, it enables the scientific community to access data which |
| 352 | would otherwise potentially be lost in time. On the rare occasions where the sampling |
| 353 | locations are presented as dots on a map (without accompanying exact geographical |
| 354 | information), the longitudes and latitudes were "hand-traced" and the approximate |
| 355 | geographical information were reported. We acknowledge that this process is |
| 356 | accompanied with uncertainties, but feel they are acceptable given the value and |
| 357 | irreproducibility of the data in older papers. |
| 358 | We believe this manual data extraction from older papers has a significant added value, |
| 359 | as for normal research projects it would not be feasible to invest this time. |

| 361 | For many of the more recent papers, the researcher could extract data from csv files or |
| 362 | paper SI Tables. |

Web crawlers (e.g., written in Python) that extract web-based data were found not precise enough to do this work. Therefore it was elected to undertake this manually by a trained researcher who is familiar with the field, methods and data types.

Thus far, we have not retrieved data from other databases. As the database grows (during the above-mentioned dedicated post-doc project), there will be an increased opportunity to do dataset by dataset comparison. In the long-term we would to link to other databases (e.g., Pangaea) to promote facile data access/exchange, but this is beyond the current scope of the project.

**Q2. What search engines and search terms (including any refinements) were used, and how were quality controls implemented?**

Search engine and search terms: The Google Scholar search engine was used within the ETH network, which allows access to nearly all journals. The search terms used were: "organic carbon in surficial surface sediments", "TOC in surficial surface sediments" and "radiocarbon/$^{14}$C in surficial surface sediments".

Quality controls: only peer-reviewed papers were used, and coordinate systems were converted where necessary to the now-widely accepted WSG84 coordinate system. The researcher was supported by Eglinton by screening datasets and looking for obvious outliers. Additionally, an automated python script checked for outlying values to provide a last external quality check by the lead author (Van der Voort).

**Q3 How many papers did the initial search yield, and how was the final subset arrived at?**

The initial search yielded several hundred papers (>300 papers). In our reply to Question 1B (Q1B, How were papers selected? Line 340) we have described how we arrived at the final subset.

**Q4 When was the database accessed?**

The most recent update of the MOSAIC website was done this January. The website always includes it's unique DOI and a timestamp of the most recent update. Users can refer to this when they use the dataset. Additionally, of course, the original paper DOI is provided for every single datapoint. We expect to update parameter space on a quasi-annual basis and add datasets in higher frequency, which will be enabled by a dedicated post-doc project starting March 1st.

**Q5 Does this database contain data from other databases?**

At the time of data collection, the data was acquired from the papers directly.

**Q6 What downstream processing of the data, or meta-data, was necessary? e.g. were units converted, how was lat and long derived/converted to the same projection, how was a position assigned to biogeographical zones etc?**

Yes, we did downstream processing, and this has now been explained more thoroughly in the text. All data was converted to the standard units (e.g., Total Organic Carbon in weight percent, $^{13}$C in permille and latitude and longitude in the WSG84 coordinate system). Whenever the sampling year was detailed in the text, the Fm, percent modern or 14C age could be converted to Delta $^{14}$C or vice versa. If data was given in percent modern carbon or 14C age, they were also converted. The assignment to biogeographical zones was done manually by the collecting researcher. For a future iteration of MOSAIC, we are working on the automatic allocation of zones (both biogeographical and EEZs) using Python, but this remains a work in progress.

Line 177-180 - this is admirable and will be beneficial, but at present does not exist. This aspiration should be omitted from the current description. Instead, the authors should add in the Data Accessibility section that updates will take place (how often? when?) and how to access the latest version of the database. I assume that each iteration will have a documented history and version number thats traceable? If not, this needs to be implemented from the outset.

Thank you for this suggestion. We have clarified the text. On the website, the main and most abundant data can be easily accessed. Using SQL, highly detailed information (e.g., cruise name, sample name or name of $^{14}$C lab) can be accessed. Due to space limitations, we cannot collapse a database with nearly a dozen tables into a user-friendly Web portal with a table. Hence, we also provide the detailed data in SQL format, and have also provided an introduction with examples on how to access specific details. We aim to update the database approximately yearly, and the version and DOI is included on the website under the "How to use this app & app version" tab. We have clarified this in the main text.

e 186 - can each individual datapoint be traced back to the individual source (paper)? It will be important that users of the data can look at the context of each datapoint by going back to the original source if necessary. in other words, is there a unique identifier that matches the data value to the specific paper from which it was extracted? This is essential and needs to be included if not already done so.

Yes, absolutely. We have clarified this in the text. All datapoints are accompanied by the DOI (last column of the table). This way, indeed, a user can go back to the original source when they're interested with just a few clicks.

Line 221 - how are submitted data quality checked? make this clear here.
Thank you for this comment, this is an important point and is detailed in the text. Briefly, an automated preliminary check is done in Python to flag suspicious values, followed by a hands-on check by an in-house expert. We have also included these details explicitly in the SI.

In more detail:
This script auto-checks the values of key parameters such as as latitude, longitude, carbon and nitrogen content, $^{13}$C, $^{14}$C, $CaCO_3$ content, $SiO_2$ content and sediment texture-related parameters. The auto-check produces a log file with flags for unexpected values. In turn, the flags point to the exact line containing possible out-of-bound values. For example, for TOC (%), if values are negative, there will be a prompt "*cannot be negative, please check*", when values are > 2 and <20 there is a prompt "*is quite high. Are you sure it is correct?*" and lastly if values are > 20 there is the prompt "*value is high. Please check units*". Each flag is accompanied by a line number to locate the possibly erroneous data. These flags then trigger a manual quality check of the data by an expert in-house user. We have now included all checks in detail in the SI.

In other words, the work is automated to a large degree, but is overseen by an expert in-house member of the ETH Biogeoscience group.

8. Line 228 - how exactly are unexpected values determined? How is this reconciled with unexpected, or outlier variables, that are nevertheless real? Need to reassure the reader that the data is not being sanitised to some pre-determined criteria or parameters.

Thank you for this question. With the automated Python-powered check, suspicious values are only flagged (with data line number, so it's easy to locate the data), not removed or deleted. This then allows for an in-house expert user to manually check the flagged data, as our experience is that – while laborious – scripts do not substitute for "human" checks for oceanographic consistency in the data. For instance, if the TOC values are high (> 2 and <20), and there is a prompt: "*is quite high. Are you sure it is correct?*" the in-house expert will have a closer look. If the values are from a zone of hypoxia or anoxia, or high biological productivity, they will understand that the data is likely correct. A common issue, for example, is that TOC data is provided in mg/g instead of percentages.

Many parameters (e.g., texture parameters % clay, silt, sand), values cannot be negative or over a hundred percent, so such values are also flagged. Isotopic values can of course be negative, but should fall within reasonable ranges.

Initially, details of this were in the python script, but to make them more accessible they have now been added in the SI.

9. Data quality control - this section needs expanding, as stated earlier, to include quality controls at the point of data collection. The current section only lists quality control post collection. In addition, this section would benefit from some explanation/justification of the detail, supported by citations where necessary/appropriate

Thank you for this comment, we have expanded and have included the details of the quality control at the data collection.

10. Section 2.3.5 - it would be beneficial for the supplementary material to include an "idiots guide" for how to complete a search and extract the data for a simple and more complex query example. For example, what are the step through processes to extract a global dataset versus just one region, or whatever is likely to be a common query. This should be made readable and accessible to users that have never used SQL or programming, or that have little or no experience of extracting data. The video is a useful addition in this regard, but a manual type addition to the supplementary material would be helpful.

Thank you for this comment, we have followed up on your suggestion and included a step-by- step introductory guide for new users on how to access the MOSAIC SQL database in the SI.

11. Section 3.1 - much of this section is unnecessary and not particularly helpful. the description of the distribution of data is only relevant to the database as it now stands, but as highlighted in the papers, the database will be updated. hence, such statements will be misleading at the point of the first update. Instead, purely descriptive statistics that relate to the database structure (i.e. not interpretative information) should be presented, such as the number of observations for each variable, categorised by region, water depth and other column headings in the database. Presently, it is hard for the reader to understand what the database contains without entering the database itself. As made above (point #1), this section morphs from being a database description to a paper thats interpreting the data. In my opinion, as interesting as the summaries are, the latter has no place here. If the authors wish to interpret the data, they should write a separate contribution and publish elsewhere

Thank you for this comment. Briefly, you stress that this paper should be descriptive, not interpretative and that sufficient details w.r.t. data points should be provided. Regarding the point of descriptive vs. interpretative, we follow the line of other ESSD database papers where it is common practice to show illustrative examples of data that can be extracted. We stress in the main text that the examples are illustrative and not definitive, and to highlight the sorts of information that are already emerging form this database. Regarding the point that sufficient details w.r.t to datapoints should be provided, we included in Table 1 an overview of main variables and their abundance, and in Figures 2 and 3 an overview of location and distribution (average, mean, median and n) of all key variables.

12. Section 3.2 - this can be condensed significantly, many of these points have been made in
the Abstract and Introduction. The text would also benefit from reaching out to other fields,
perhaps offering other areas that these data may be relevant to that have not received attention
previously.

Thank you for these comments. Thanks to your review, it has come to our attention that also
MOSAIC may also have relevance for the field of seafloor ecology. We have now included
some text on this point as well and citations to the papers by e.g. Isla and DeMaster, (2018),
Snelgrove et al., (2018), and of course Middelburg, (2018). Furthermore, as suggested, we have
also condensed this section.

13. Section 3.3 - this section is quite weak and not very compelling. It is not entirely clear
whether (i) the data contained in this database is a subset of the other databases mentioned, (ii)
how these data differ from other inventories and what the pros and cons of these data are in
relation to specific areas of research (maybe include reference to other databases that may form
good companions to these data), (iii) and why a user should opt for using these data? Some
aspects of these matters are listed, but only in very general terms that lack specifics. Much
more explicit arguments need to be made here.

Thank you for your comment. We have addressed the comments as detailed below:
- **Q13- I the data contained in this database is a subset of the other databases**
**mentioned**
Thank you, as per your suggestion we have expanded details on data collections so this
has been clarified. MOSAIC has been created by collecting the data from > 200 paper
publications, numerous from which data could only extracted manually from PDFs.

- **Q13- II how these data differ from other inventories and what the pros and cons**
**of these data are in relation to specific areas of research (maybe include reference**
**to other databases that may form good companions to these data)**
We have explained how MOSAIC differs from other initiatives and have also included
the references to other databases.
In the section 3.3 we describe that MOSAIC differs from these and other initiatives in
its targeted approach with a primary focus on (*i*) pro-actively collating data pertinent to

OC burial on continental margins, (*ii*) upper sediment layers (nominally < ~ 1m) that encompass early diagenetic processes and recent deposition (as opposed to down-core studies that seek to reconstruct past ocean and climate conditions), and (*iii*) radiocarbon information that bridges to equivalent databases for other carbon cycle compartments. In this way, we envision that it will serve as a resource to enable "on-stop shopping" for biogeochemical and sedimentological information on continental margin surficial sediments. While thus far data ingested into MOSAIC has been retrieved from the primary research literature, future efforts will focus on harmonizing and linking with other databases in order to improve overall connectivity of information.

- **Q13 – III (iii) and why a user should opt for using these data? Some aspects of these matters are listed, but only in very general terms that lack specifics. Much more explicit arguments need to be made here.**

- Thank you for this comment, we have added specifics in section 3.3. We provide a user-friendly interface which is very transparent, where sample location and data source are directly provided (all DOIs are provided). MOSAIC constitutes that largest collection of ocean shelf sedimentary data in this format thusfar.

- We anticipate that MOSAIC will serve as a key research and teaching resource for biogeochemists focusing on contemporary biogeochemical processes as well as seeking to interrogate sedimentary archives to develop records of past oceanographic conditions.

14. Section 4 - add a sentence that states what version of the database this paper is referring to/describing, and how often users can expect updates to the database (e.g. periodically, annually?). I suggest it will also be advantageous to state how errors can be reported.

Thank you, we have now included this. Both new data and errors or bugs can be reported to mosaic@erdw.ethz.ch. We expect to do a semi-annual update, enabled by a fully funded post-doc in the Eglinton group dedicated to MOSAIC starting March 2021.

Section 5 - this section is repetitive of the sections above and does not add anything new. This section needs revising to pick up from where the Introduction left off.

*Thank you. We have shortened the section and revised it to pick up where the Introduction left off.*

16. Table 1- the database contains 8706 entries with latitude and longitude, but only about half of these have a water depth associated with them - could those that do not have a water depth be estimated using, for example, Google earth based on the lat and long co-ordinates? I note the comment re GEBCO, but the same comment made earlier about the state of the database at the point of publication versus aspirations stands.

*This is correct. We have looked into connecting to the GEBCO database to also see if we can auto-query the depth using the lat and lon and include this, but this is technically not trivial to do. In the present iteration, the depth is not available for all but nonetheless numerous datapoints, as clearly stated. Users can, of course, look up the depth information with GEBCO using the lat and lon they get from their data.*

Overall, I am supportive of the communication, but as the manuscript now stands it does not include sufficient detail about how the data were derived and forms an incompatible mix of existing versus aspirational database properties. I would see both of these as moderate revisions.

*Thank you for your support. We have included additional details w.r.t. how the data was derived ad we have clarified the contents of the current iteration, and potential for growth (the ocean is vastly expansive and complex!).*

**Referee #3 Paula Reimer**

The MOSAIC database will be an extremely valuable tool for research in marine carbon storage. It is easily used and expandable. I am not an expert in marine sediments but I was glad to see the authors are thinking about adding on 14C of sediment fraction from density separations or thermal decomposition which can contribute important information on carbon source which of course would require fields for the density or temperatures used for each 14C measurement. I found the examples useful though obviously more data is needed. I definitely support publication of the MOSAIC article in ESSD after minor revisions.

Dear Prof. Reimer, Paula,

Thank you very much for your encouragement and positive feedback! It is very much appreciated. We have incorporated all suggested revisions.

I have only a few very minor comments/corrections:

Line 73: Why reference Reimer et al. 2009 when the 14C archives have been updated in Reimer et al 2020 (doi: 10.1017/RDC.2020.41)? Thank you - improved

Line 159-161: 'Sediment depth profile data primarily used to examine diagenetic profiles, and to constrain sedimentation rates, mixed layer depths, redox gradients, as well as to determine carbon fluxes and inventories'. This is an incomplete sentence. Thank you – indeed it was, this has been corrected.

Line 176-177: 'and the stable carbon isotopic composition (d13C and 14C values) of OC'. Delete 'stable' since 14C is not stable. Thank you – corrected.

Line 199: Presumably $\Delta 14C$ is not age corrected but, in any case, the date of the 14C measurement would be needed to convert to Fm. If it is age corrected then the year of collection is also needed.

Thank you, indeed, and whenever this possible this has been done. We have clarified this in the text.

Line 327: 'coupled the application. . .' Presumably this should be 'coupled with' Thank you – corrected.

Line 345: '14C' should have a superscript 14C Thank you – corrected.

References

Arndt, S., Jørgensen, B. B., LaRowe, D. E., Middelburg, J. J., Pancost, R. D. and Regnier, P.: Quantifying the degradation of organic matter in marine sediments: A review and synthesis, Earth-Science Rev., 123, 53–86, doi:10.1016/j.earscirev.2013.02.008, 2013.

Avelar, S., van der Voort, T. S. and Eglinton, T. I.: Relevance of carbon stocks of marine sediments for national greenhouse gas inventories of maritime nations, Carbon Balance Manag., 12(1), 10, doi:10.1186/s13021-017-0077-x, 2017.

Bianchi, T. S.: The role of terrestrially derived organic carbon in the coastal ocean: A changing paradigm and the priming effect, Proc. Natl. Acad. Sci., 108(49), 19473–19481, doi:10.1073/pnas.1017982108, 2011.

Griffith, D. R., Martin, W. R. and Eglinton, T. I.: The radiocarbon age of organic carbon in marine surface sediments, Geochim. Cosmochim. Acta, 74(23), 6788–6800 [online] Available from: http://linkinghub.elsevier.com/retrieve/pii/S001670371000493X (Accessed 2 August 2013), 2010.

Isla, E. and DeMaster, D. J.: Labile organic carbon dynamics in continental shelf sediments after the recent collapse of the Larsen ice shelves off the eastern Antarctic Peninsula: A radiochemical approach, Geochim. Cosmochim. Acta, 242, 34–50, doi:10.1016/j.gca.2018.08.011, 2018.

Middelburg, J. J.: Reviews and syntheses: to the bottom of carbon processing at the seafloor, Biogeosciences, 15(2), 413–427, doi:10.5194/bg-15-413-2018, 2018.

Snelgrove, P. V. R., Soetaert, K., Solan, M., Thrush, S., Wei, C. L., Danovaro, R., Fulweiler, R. W., Kitazato, H., Ingole, B., Norkko, A., Parkes, R. J. and Volkenborn, N.: Global Carbon Cycling on a Heterogeneous Seafloor, Trends Ecol. Evol., 33(2), 96–105, doi:10.1016/j.tree.2017.11.004, 2018.

Voort, T. S. Van Der, Mannu, U. and Blattmann, T. M.: Deconvolving the fate of carbon in coastal sediments, Geophys. Res. Lett., 45(June), 4134–4142, doi:10.1029/2018GL077009, 2018.

---

## Author Response (AR2)

Dear David,

Thank you very much for your positive reaction! We have implemented all suggested corrections and look forward to sharing MOSAIC with the community.

On behalf of all my co-authors,

Tessa
* * *
Fm (Fraction modern) used frequently but not defined until legend to Fig 5 (line 458). → corrected, thank you

Line 197 - "includes is an"? → corrected, thank you

Lines 300 to 303, confusion. del13C values show two distinct peaks, heavier marine vs lighter (more depleted) terrestrial, but you provide a single mean, median? E.g. Fig 3.b., in my training this distribution would not qualify as a normal (Gaussian) distribution. →corrected, thank you. Indeed, as the text indicated there are two distinct peaks, and a single mean does not accurately represent the distribution. We have now provided the two modes of the bimodal distribution, indicative of the aquatic and terrestrial sources of carbon.

Supplement line 72: repsecite? → corrected, thank you
Supplement contains section S3 twice? → corrected, thank you, all SI references have been double-checked.